# Controllable Synthesis and Growth Mechanism of Interlayer-Coupled Multilayer Graphene

**DOI:** 10.3390/nano13192634

**Published:** 2023-09-25

**Authors:** Xudong Xue, Mengya Liu, Xiahong Zhou, Shan Liu, Liping Wang, Gui Yu

**Affiliations:** 1Beijing National Laboratory for Molecular Sciences, CAS Research/Education Center for Excellence in Molecular Sciences, Institute of Chemistry, Chinese Academy of Sciences, Beijing 100190, China; xuexudong@hebeu.edu.cn (X.X.); b20200192@xs.ustb.edu.cn (M.L.); zhouxiahong@iccas.ac.cn (X.Z.); liushan@iccas.ac.cn (S.L.); 2School of Materials Science and Engineering, University of Science and Technology Beijing, Beijing 100083, China; lpwang@mater.ustb.edu.cn; 3School of Chemical Sciences, University of Chinese Academy of Sciences, Beijing 100049, China

**Keywords:** graphene, multilayer graphene, CVD

## Abstract

The potential applications of multilayer graphene in many fields, such as superconductivity and thermal conductivity, continue to emerge. However, there are still many problems in the growth mechanism of multilayer graphene. In this paper, a simple control strategy for the preparation of interlayer-coupled multilayer graphene on a liquid Cu substrate was developed. By adjusting the flow rate of a carrier gas in the CVD system, the effect for finely controlling the carbon source supply was achieved. Therefore, the carbon could diffuse from the edge of the single-layer graphene to underneath the layer of graphene and then interlayer-coupled multilayer graphene with different shapes were prepared. Through a variety of characterization methods, it was determined that the stacked mode of interlayer-coupled multilayer graphene conformed to AB-stacking structure. The small multilayer graphene domains stacked under single-layer graphene was first found, and the growth process and growth mechanism of interlayer-coupled multilayer graphene with winged and umbrella shapes were studied, respectively. This study reveals the growth mechanism of multilayer graphene grown by using a carbon source through edge diffusion, paving the way for the controllable preparation of multilayer graphene on a liquid Cu surface.

## 1. Introduction

Graphene has great application prospects in electronics, optics, and biology [1,2,3,4]. The performance of electronical graphene is strongly dependent on the layer number and stacking structure. For instance, single-layer graphene exhibits an extremely high carrier mobility. However, it is a zero-bandgap semimetal, making it difficult to integrate with current popular semiconductor processes. The properties of bilayer graphene undergo significant changes compared to single-layer graphene, leading to outstanding performances in various frontier fields. For example, the bandgap of AB-stacked bilayer graphene can be opened under a perpendicular electric field. “Magic-angle” graphene demonstrates superconductivity, and unconventional conduction phenomena occur in small-angle, twisted, bilayer graphene [5,6,7]. Additionally, trilayer graphene also holds great potential in the fields of superconductivity and thermal conductivity [8,9]. In addition to the well-known excellent physical properties of monolayer, bilayer, and trilayer graphene, multilayer graphene also holds enormous potential for various applications. For instance, multilayer graphene exhibits promising performance in transparent conductive films, thermal conductivity, superlubricity, and flexible devices [10,11,12,13].

In recent years, with the advancement of graphene fabrication techniques, the technology for preparing large-area single-layer graphene using the chemical vapor deposition (CVD) method has become increasingly mature. Research efforts have extended to the fabrication of bilayer and multilayer graphene [14]. For the preparation of bilayer graphene, the CVD method has seen rapid development in fabricating common stacking configurations of bilayer graphene (excluding small-angle twisted bilayer graphene) [9]. However, exploration is still further needed in the CVD fabrication of large-area, high-quality trilayer and multilayer graphene, as well as in studying their stacking structures and growth mechanisms. With the continuous study in CVD-derived graphene on Cu surfaces, the mode of “self-limiting” growth of monolayer graphene on Cu surfaces has been gradually broken. Research has shown that multilayer graphene grown on Cu surfaces adopts an “inverted cake” structure [15,16]. Yan et al. succeeded in preparing multilayer graphene on Cu foil under high-pressure, long-duration annealing conditions [17]. Deng et al. further discovered that the presence of Cu steps during the annealing process on a Cu substrate allowed carbon sources to diffuse beneath the first layer of graphene, providing a possibility for the fabrication of multilayer graphene [18]. Additionally, Nguyen et al. achieved the growth of four-layer graphene by preparing high-quality Cu–Si substrates and subsequently growing graphene on them [19]. Recently, the epitaxial growth of multilayer graphene films has been achieved on single-crystal Ni (111) surfaces through uniform segregation from a solid carbon source [20]. The latest research has reported the formation of onion-like multilayer graphene and explained its mechanism of growth along spin-spiral lines [21]. However, there is still a need for extensive scientific research on the formation and growth processes of multilayer graphene, especially on different substrates. For liquid Cu substrates, the preparation of hierarchical and umbrella-like structured graphene domains on liquid Cu substrates has been reported [22,23]. However, the formation and growth process of interlayer-coupled multilayer graphene crystals are not fully understood yet, and there are many questions that require further investigation.

In this study, we developed a simple method and control strategy for preparing interlayer-coupled multilayer graphene based on a liquid Cu substrate. By adjusting the gas flow rate in the system, we achieved precise control over the carbon source supply, resulting in the fabrication of interlayer-coupled multilayer graphene with different morphologies. We conducted the first investigation and characterization of morphology and structure of winged interlayer-coupled multilayer graphene. By using characterization techniques, such as SEM, Raman spectroscopy, AFM, and TEM, we confirmed that this type of multilayer graphene exhibits an AB-stacked structure. We also observed the phenomenon of small multilayer graphene stacking beneath single-layer graphene and studied the growth process and mechanism of interlayer-coupled multilayer graphene with different morphologies. This method demonstrates the significant potential of using a liquid Cu surface CVD for the controlled synthesis of multilayer graphene.

## 2. Materials and Methods

To ensure the cleanliness of the Cu foils (100 μm thick, 99.8% purity) and W foils (100 μm thick, 99.95% purity) prior to graphene growth, a series of cleaning steps were performed. First, the foils were immersed in diluted hydrochloric acid, followed by rinsing with deionized water, ethanol, and acetone using ultrasonication. Subsequently, the Cu/W foil was placed inside a quartz tube positioned in the heating zone of a Lindberg/Blue M TF55035A furnace. The growth of graphene was carried out by using atmospheric pressure chemical vapor deposition (CVD) with a mixture of gases, including methane (CH_4_, 99.9% purity), hydrogen (H_2_, produced by a hydrogen generator), argon (Ar, 99.999% purity), and diluted oxygen (5%). Before the graphene growth process, Ar gas was flushed into the quartz tube to remove any trapped air in the system. The furnace was then heated to 1170 °C over a period of 60 min and annealed for 30 min with a flow rate of 100 standard cubic centimeters per minute (sccm) of H_2_. During the graphene growth stage, specific flow rates of CH_4_ and Ar gases were introduced into the system to facilitate graphene growth, while the flow rate of hydrogen was adjusted accordingly. Once the growth step was completed, the CH_4_ gas was switched off, and the furnace was rapidly cooled down to room temperature.

After the graphene was grown on a liquid Cu substrate, it was transferred to a SiO_2_/Si substrate by using an electrochemical delamination method [24]. Initially, the graphene samples on the Cu substrate were protected by a solidified PMMA (polymethyl methacrylate) layer (4000 r/min). Subsequently, the PMMA/graphene film was electrochemically separated from the Cu substrate by bubbling H_2_ in a 1 M NaOH aqueous solution. Once the PMMA/graphene was completely detached from the Cu substrate, it was transferred onto a 300 nm thick SiO_2_/Si substrate or Cu grids and left to dry naturally. Finally, prior to characterization, the dried PMMA/graphene film was immersed in acetone at 60 °C to dissolve the PMMA layer.

## 3. Results

Owing to the low solubility of carbon in Cu, it is easier to grow monolayer graphene on Cu substrates [25]. Similarly, on a liquid Cu substrate, monolayer graphene can also be easily grown [26]. Compared to solid Cu, the surface of liquid Cu is smoother, making it difficult to simultaneously form multiple nucleation sites for growing multilayer graphene. Therefore, to prepare multilayer graphene, it is necessary to break this “self-limiting” growth mode and provide measures to promote the simultaneous growth of multilayer graphene at the same nucleation site or the continuous growth of multilayer graphene on pre-existing monolayer graphene. The growth temperature required for graphene growth on liquid Cu substrates is higher than that on solid Cu substrates. This high-temperature condition provides more energy during the growth process, resulting in faster diffusion of carbon species and accelerated graphene growth on the liquid Cu surface [27]. The rapid diffusion of carbon species can easily lead to their accumulation and the formation of new nucleation sites. Subsequently, graphene continues to grow and forms multilayer graphene structures. Under high-temperature conditions, providing a greater amount of carbon than what is required for monolayer graphene growth allows for the rapid enrichment of carbon species, creating the necessary conditions for the growth of multilayer graphene. The graphene single-crystals grown on the Cu surface exhibit an “inverted cake” structure [28]. To better illustrate the preparation of graphene, an inverted method was used, where the sequentially grown multilayer graphene beneath the first layer was inverted and depicted on top of the first layer of graphene, with different colors representing different layers. As shown in Appendix A, the carbon source begins to enrich from the edge and then forms small interlayer-coupled multilayer graphene, and the interlayer-coupled multilayer graphene gradually grows with time.

The interlayer-coupled multilayer graphene samples prepared on liquid Cu were observed under SEM to examine their morphological features. As shown in Figure 1a, at the lower right corner of the hexagonal graphene domain (indicated by a white dashed box), multiple inverted cake-like hexagonal layers were observed. These layers have a darker color compared to the outermost layer of monolayer graphene, indicating a higher number of graphene layers in this region [23]. Continuing along the direction indicated by the red arrow in the figure, the colors of the small hexagonal graphene structures gradually darken, indicating an increasing thickness as the hexagonal graphene layers stack. Additionally, at the center of the interlayer-coupled multilayer graphene, which corresponds to the nucleation site, there is an extension in the form of a black triangular tail. The deepest color in this region suggests the highest number of layers. The schematic diagram of the interlayer-coupled multilayer graphene (Figure 1b) provides a clearer representation of its structural features. The tip of the triangular tail coincides precisely with the nucleation site of the multilayer graphene, where inverted cake-like islands of interlayer-coupled multilayer graphene continue to grow. Research indicates that these inverted cake-like graphene islands form beneath the first layer of graphene [29]. It is noteworthy that the center point of the hexagonal interlayer-coupled multilayer graphene shown in Figure 1b does not coincide with the center point of the outermost large single-crystal graphene. This indicates that the interlayer-coupled multilayer graphene and the large single-crystal graphene do not originate from the same nucleation site. The large single-crystal graphene has a separate nucleation site, while the remaining interlayer-coupled multilayer graphene shares a common nucleation site. This novel type of multilayer graphene with a tail structure has not been reported before, suggesting the involvement of some novel growth mechanisms. Subsequently, the graphene samples transferred onto SiO_2_/Si were characterized by using Raman spectroscopy to determine their layer numbers and quality. Raman spectroscopy measurements were performed layer by layer along the red arrow direction in Appendix A. It was found that the outermost layer of the interlayer-coupled multilayer graphene exhibited a Raman spectrum with an *I*_2D_/*I*_G_ value of approximately 2, indicating it was a monolayer graphene [26]. The red Raman spectrum curve in Appendix A had an *I*_2D_/*I*_G_ value of approximately 1. Furthermore, considering the nearly parallel alignment of the second layer of graphene and the outermost layer of graphene at the edges (indicated by the red dashed line in Appendix A), it suggests that the second layer corresponds to an AB-stacked bilayer graphene [30]. For the incrementally layered multilayer graphene, the intensity of the 2D peak is smaller than that of the D peak. Owing to the variations in the electronic structure of graphene, the process of graphene’s double-resonance effect is affected, leading to a broadening of the 2D peak [31]. Additionally, no significant signal of the defect peak (D peak) was observed, indicating that the grown interlayer-coupled multilayer graphene exhibits high quality.

To further validate the layer variation and uniformity of the interlayer-coupled multilayer graphene, Raman mapping was performed at the location indicated by the red dashed box in Figure 2a. Raman mapping of the graphene G peak reveals the presence of the stacked interlayer coupling phenomena (Figure 2b). From the Raman mapping of the 2D peak (Figure 2c), the contour appearance of the second-layer graphene was observed, while the layer variation of the other layers was not significant. This may be attributed to the broadening of the 2D peak with increasing layers, while its intensity remains unchanged. Furthermore, Raman mapping of *I*_2D_/*I*_G_ (Figure 2d) clearly shows that the outermost layer is single-layer graphene, followed by an increasing layer count in the direction of the nucleation point, with the middle black region having the highest number of graphene layers. Raman mapping of the D peak and *I*_D_/*I*_G_ ratio (Figure 2e,f) shows low signal intensity, indicating a high quality of the entire graphene sample, which is consistent with the results from Raman spectroscopy.

To directly observe the morphology and layer number of the graphene, the graphene sample transferred onto SiO_2_/Si was examined using atomic force microscopy (AFM). As shown in Appendix A, the region of interlayer-coupled multilayer graphene with tail-like structures is clearly visible. The center position exhibits the highest height, followed by the extension of two tail structures, also positioned relatively high. Subsequently, the height variation of the graphene at the white dashed line position in Appendix A was measured, as shown in Appendix A. Because the thickness of the graphene was too large, only a decreasing trend in height from the highest nucleation point toward the single-layer graphene direction can be observed, but the layer-by-layer morphology cannot be distinguished. Further visualization of the interlayer-coupled multilayer graphene structure can be achieved through the AFM 3D image. Appendix A reveals that the height of the graphene gradually increases from the position of single-layer graphene toward the central nucleation point, and at the highest nucleation point, the interlayer-coupled graphene morphology radiates outward.

To microscopically characterize the layer number of the interlayer-coupled multilayer graphene, the grown graphene sample was transferred onto a coordinate grid for subsequent TEM characterization. As shown in Figure 3a, the hexagonal graphene domain exhibited an additional tail. Because it was challenging to distinguish the layer number of the graphene, a schematic diagram (Figure 3b) was used for illustration. After locating the interlayer-coupled multilayer graphene on the microgrid, TEM imaging and selected-area electron diffraction (SAED) characterization were performed. Along the direction indicated by the red arrow in Figure 3a, four positions were selected for SAED tests, and the results are shown in Figure 3c–f. The diffraction patterns at these four positions had the same orientation, exhibiting a set of hexagonal diffraction patterns with six-fold symmetry. With the exclusion of uniform single-layer graphene, this indicates that there is no mutual twisting between different layers of the interlayer-coupled multilayer graphene, and they adopt an AB stacking arrangement [15]. Subsequently, the layer number variation was analyzed by examining the diffraction intensities of the respective diffraction patterns. As shown in the inset of Figure 3c, the intensity of the outer-ring diffraction point *I*_(1-210)_ is lower than that of the inner-ring diffraction point *I*_(1-100),_ indicating single-layer graphene. Continuing to observe the diffraction intensity insets in Figure 3d–f, the intensity of the outer-ring diffraction point *I*_(1-210)_ is higher than that of the inner-ring diffraction point *I*_(1-100)_, suggesting the presence of bilayer or multilayer graphene. This result is consistent with the schematic diagram of the interlayer-coupled multilayer graphene with a tail (Figure 3b). Further analysis was conducted by examining the edges of the graphene to determine its layer number. In Figure 3g, it was observed that the edge of the graphene corresponded to a single layer (corresponding to the position indicated by the green line in Figure 3a and the diffraction pattern of a single layer in Figure 3c) [24]. From Figure 3h, it was observed that the edge of the graphene corresponded to a multilayer structure (corresponding to the position indicated by the pink line in Figure 3a and the diffraction pattern of multiple layers in Figure 3f) [9,15]. In short, TEM analysis confirmed that the structure consisted of single-layer edges and locally stacked AB-stacked multilayer interlayer-coupled graphene.

## 4. Discussion

Upon observation, it was found that this interlayer-coupled multilayer graphene exhibited various morphologies. These included the structure with a tail (Appendix A), the structure with multiple tails (Appendix A), and the umbrella-like structure (Appendix A). In order to highlight the novelty of the prepared multilayer graphene morphologies and preparation methods, we summarized the results and compared that with previous works (Appendix A). The multilayer graphene domains have promising performance in transparent conductive films, thermal conductivity, superlubricity, and flexible devices [10,11,12,13]. Moreover, the regulation of experimental conditions might lead to the realization of twisted multilayer graphene, which also has important application prospects in the field of superconductivity [32].

The presence of different morphologies in interlayer-coupled graphene suggests the formation of multilayer graphene with varying thicknesses in different regions of the graphene. This variation in morphology could be attributed to the carbon source supply during the growth process. Owing to the limited range of control over the methane flow during graphene growth, attempts were made to indirectly regulate the carbon source supply in the actual growth process by adjusting the hydrogen gas flow rate [33]. We fixed the flow rate of the carbon source at 0.8 sccm and adjusted other parameters to observe the morphology changes in graphene. When the hydrogen gas flow rate is high (120 sccm), it inhibits the nucleation density of graphene and slows down the growth rate [34]. As shown in Appendix A, the graphene domains are primarily single-layer graphene, and the presence of interlayer-coupled multilayer graphene was not directly observed. When the hydrogen gas flow rate is reduced to 100 sccm, occasional structures of interlayer-coupled multilayer graphene were observed (Appendix A). As the hydrogen gas flow rate further decreases to 80 sccm, the proportion of interlayer-coupled graphene domains increases (Appendix A). When the hydrogen gas flow rate continues to decrease to 60 sccm, an excess of carbon source during the growth process leads to the formation of large-area interlayer-coupled multilayer graphene domains (Appendix A). Throughout the modulation of the hydrogen gas flow rate, it is difficult to observe large areas of interlayer-coupled multilayer graphene with tails. Therefore, solely adjusting the hydrogen gas flow rate may have limited effects on achieving significant changes in the interlayer stacking morphology of graphene over a large scale.

Further investigation was conducted by controlling the carrier gas flow rate to study the effect of gas flow on the preparation of interlayer-coupled multilayer graphene. As shown in Figure 4a, when the Ar gas flow rate is relatively low (300 sccm), the carbon source supply is in excess, resulting in the formation of nonuniform onion-like multilayer graphene [21,35]. When the Ar gas flow rate is increased (500 sccm), the proportion of Ar gas also increases, leading to a reduction in the actual carbon source supply during graphene growth. As a result, the nonuniform layered onion-like graphene disappears, and single-layer graphene coexists with umbrella-shaped interlayer-coupled multilayer graphene (Figure 4b) [36]. When the Ar gas flow rate continues to increase (700 sccm), a decrease in the proportion of umbrella-shaped graphene is observed, and interlayer-coupled multilayer graphene with tails starts to appear (Figure 4c). Furthermore, when the Ar gas flow rate is further increased (900 sccm), the emergence of large areas of interlayer-coupled multilayer graphene with tails is observed (Figure 4d). This may be attributed to a dynamic equilibrium growth state achieved by diluting the actual carbon source supply under the combined effect of an increased gas flow velocity and an increased Ar gas proportion.

Subsequently, we characterized the umbrella-shaped interlayer-coupled multilayer graphene. As shown in Appendix A, the darker-colored hexagonal regions exhibit the structure of umbrella-shaped interlayer-coupled multilayer graphene. After transferring it onto a SiO_2_/Si substrate for further observation, we found that the domains of umbrella-shaped interlayer-coupled multilayer graphene appear significantly darker in color under optical microscopy compared to single-layer graphene (Appendix A). Based on the previous analysis, it is known that interlayer-coupled graphene with tails is a multilayer structure; therefore, the overall structure of this umbrella-shaped interlayer-coupled multilayer graphene should also be multilayered. Raman spectroscopy (Appendix A) confirmed that the umbrella-shaped interlayer-coupled multilayer graphene indeed possesses a multilayer structure [37]. Additionally, the absence or minimal presence of the D peak signal in the Raman spectrum indicates the high quality of the graphene.

Raman mapping tests were conducted within the red dashed box region shown in Appendix A to determine the layer number and uniformity of the umbrella-shaped interlayer-coupled multilayer graphene domains. From the Raman mapping spectra of the G peak (Appendix A) and 2D peak (Appendix A), it can be observed that the umbrella-shaped interlayer-coupled multilayer graphene exhibits good uniformity. The Raman mapping results of the *I*_2D_/*I*_G_ ratio (Appendix A) indicate that the entire region consists of a multilayer morphology. Furthermore, the slightly stronger signal of the D peak (Appendix A) suggests the presence of defects in this hexagonal multilayer graphene. However, the signal intensity in the *I*_D_/*I*_G_ Raman mapping map (Appendix A) of the graphene region is low, indicating that the quality of the prepared umbrella-shaped interlayer-coupled graphene is still high. TEM tests were conducted on the umbrella-shaped multilayer graphene transferred onto the microgrid to observe its crystallinity. From Appendix A, it can be observed that the transferred graphene maintains good integrity in terms of morphology. In low-magnification TEM observations, the graphene edge appears thick (Appendix A). At higher magnification, the graphene edge reveals multiple layers, approximately 10 layers (Appendix A). Subsequently, SAED characterization was performed at different positions of the graphene domain. The results show that the diffraction patterns of the graphene exhibit the same pattern and consistent orientation (Appendix A–f). Therefore, it can be confirmed that the graphene is interlayer-coupled multilayer graphene with AB-stacking structure [23,36].

Although the tail regions of winged-shaped and umbrella-shaped multilayer graphene appear uniform under an optical microscope and SAED patterns in TEM exhibit the AB-stacking structure for both morphologies, a careful examination of the intermediate states of graphene morphology reveals the formation of various regions with different shades beneath the single-layer graphene (Figure 5a). Further Raman mapping was performed on the region outlined by the white dashed box in Figure 5a. From the G peak Raman mapping (Figure 5b), it is evident that the signal intensity distribution across the entire image is uneven, indicating nonuniform layer thickness of the graphene. Additionally, in the *I*_2D_/*I*_G_ Raman mapping (Figure 5c), the outer contour of the graphene corresponds to single-layer graphene, with an *I*_2D_/*I*_G_ value greater than 2. However, the *I*_2D_/*I*_G_ values for the inner regions of interlayer-coupled small multilayer graphene are less than 2, indicating that the graphene sample is multilayer in structure. However, from the *I*_D_/*I*_G_ Raman mapping (Figure 5d), it can be observed that despite the presence of many small multilayer graphene islands, the overall quality of the graphene obtained is still relatively high. Based on the above analysis, it can be concluded that during the growth process of graphene, numerous small multilayer graphene structures are formed beneath the single-layer graphene.

Further observations reveal that during the growth process of graphene, when there is an excess of carbon species, carbon begins to accumulate at the edges of single-layer graphene [21]. As shown in Appendix A, darker regions start to appear at the edges of the single-layer graphene. Raman mapping was conducted on this region, and from the G and 2D peak Raman mappings (Appendix A), it is evident that the edges of the graphene exhibit regions with nonuniform layer thickness. The *I*_2D_/*I*_G_ Raman mapping more clearly shows the presence of small multilayer graphene domains with nonuniform layer thickness at the edges of the single-layer graphene (Appendix A). Although these small multilayer structures aggregated at the edges of the single-layer graphene exhibit higher defect density compared to the single-layer graphene (Appendix A), the overall quality of the graphene in this region appears to be acceptable based on the *I*_D_/*I*_G_ Raman mapping results (Appendix A). As the accumulation of carbon species at the edges of graphene increases, the accumulated carbon species start to gradually diffuse from the edges into the single-layer graphene. In Appendix A, it can be observed that the edges of the single-crystal graphene exhibit the morphology of a layered graphene structure with wing-like features. The Raman mapping of the G peak and *I*_2D_/*I*_G_ (Appendix A) also indicate the formation of a small region of multilayer graphene in this area, which then continues to grow into the interior of the single-layer graphene.

Finally, under suitable growth conditions, the formation of winged interlayer-coupled multilayer graphene structures is achieved. As shown in Appendix A, it can be observed that multiple-winged interlayer-coupled multilayer also can be formed within the same graphene domain. We will combine the characterization results to provide a comprehensive understanding of the formation mechanisms of winged interlayer-coupled multilayer graphene and umbrella-like interlayer-coupled multilayer graphene structures. The growth mechanism of interlayer-coupled multilayer graphene with tail-like structures can be illustrated using Appendix A. When graphene is grown with an appropriate supply of carbon source, single-layer graphene can be successfully prepared. However, when the supply of the carbon source exceeds the suitable value, it disrupts the equilibrium growth state. As a result, carbon species rapidly accumulate at the edges of the single-layer graphene [21]. Under the influence of high-speed gas flow, carbon species begin to diffuse into the interior of the single-layer graphene. However, at this point, the accumulation of carbon species is insufficient to support the continuation of the growth of a second layer of graphene at the nucleation center. Therefore, the carbon species within the diffusion path anchor to a point inside the single-crystal graphene and start to grow. Owing to the excess carbon species in this region and the rapid influx of previously accumulated carbon species at the edge of graphene, a structure of interlayer-coupled multilayer graphene with tail-like structures is formed. According to the Wulff structure model of growth kinetics, the armchair edges of rapidly growing graphene will disappear, resulting in the characteristic termination of regular hexagonal graphene with zigzag edges [29]. Furthermore, the AB-stacked mode has lower energy, so the resulting multilayer graphene tends to form AB-stacking structure [18]. In the case of the wing-like AB-stacked graphene, the nucleation point does not share the same nucleation point with the single-layer graphene. This may provide new insights for controlling twisted multilayer graphene [32].

The growth mechanism of umbrella-like, interlayer-coupled multilayer graphene slightly differs from that of wing-like interlayer-coupled multilayer graphene. As shown in Figure 6a–c, due to an excessive supply of carbon source, the carbon source diffuses more vigorously toward the central region underneath the single-layer graphene. It starts to diffuse along the graphene skeleton, easily reaching the nucleation sites of the growing single-layer graphene (Figure 6a). It then continues to converge along the skeleton, forming small multilayer graphene domains. Once the diagonal skeleton lines are filled, small, multilayer graphene starts to grow along the preferred direction (Figure 6b). This may be due to the relatively heavy nature of multilayer graphene, which can sink to some extent into the liquid Cu surface. The graphene domains experience compression from surrounding Cu atoms, forming microgrooves, and allowing the small multilayer graphene to converge along the optimal growth direction [36]. Underneath the single-layer graphene, the small interlayer-coupled multilayer graphene domains continue to converge, eventually forming umbrella-like interlayer-coupled multilayer graphene (Figure 6c). The growth schematic is depicted in Figure 6d–f. This helps to provide a clearer understanding of the growth mechanism of umbrella-like interlayer-coupled multilayer graphene and lays the theoretical foundation for the further large-scale synthesis of umbrella-like multilayer graphene.

## 5. Conclusions

We developed a simple method for synthesizing interlayer-coupled multilayer graphene on a liquid Cu substrate under atmospheric pressure. By controlling the Ar gas flow at high temperatures, the “self-limiting” growth mode of single-layer graphene on the Cu surface was broken, resulting in a novel structure of interlayer-coupled multilayer graphene with wing-like features. The morphology characteristics were characterized using SEM, AFM, and optical microscopy, while the layer number and crystallinity were analyzed using Raman spectroscopy and TEM. Additionally, we discovered intermediate states in the growth process of interlayer-coupled multilayer graphene, laying the foundation for further exploration of the growth mechanism of multilayer graphene. This is an exciting development in the field of graphene synthesis.

## Figures and Tables

**Figure 1 nanomaterials-13-02634-f001:**
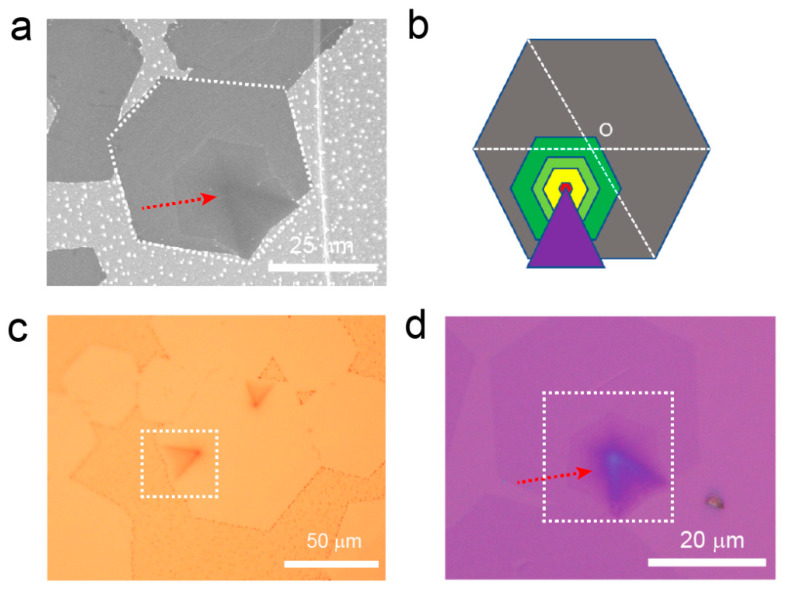
(**a**) SEM image of the interlayer-coupled multilayer graphene grown on liquid Cu surface. (**b**) Schematic diagram of the interlayer-coupled multilayer graphene. (**c**) Optical photograph of the interlayer-coupled multilayer graphene grown on liquid Cu surface after oxidation. (**d**) Optical photograph of the interlayer-coupled multilayer graphene transferred onto SiO_2_/Si substrate.

**Figure 2 nanomaterials-13-02634-f002:**
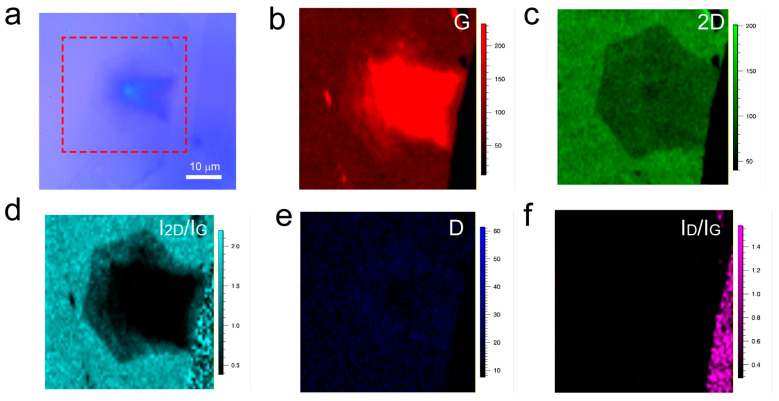
(**a**) Optical photograph of interlayer-coupled multilayer graphene. (**b**–**f**) Raman mapping of interlayer-coupled multilayer graphene.

**Figure 3 nanomaterials-13-02634-f003:**
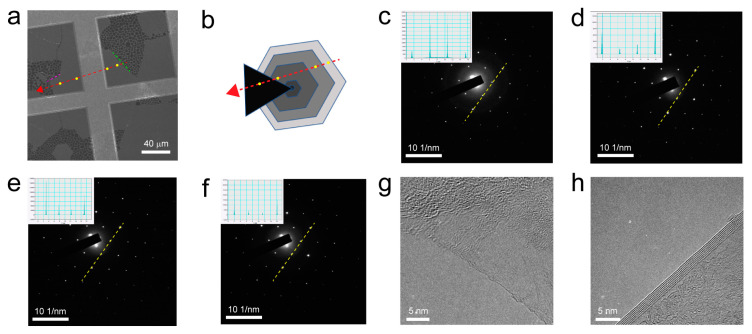
(**a**) SEM image of interlayer-coupled multilayer graphene transferred onto a Cu grid. (**b**) Schematic representation of the corresponding image in (**a**) showing interlayer-coupled multilayer graphene. (**c**–**f**) SAED patterns at four yellow dot positions along the red arrow direction in (**a**). Insets show corresponding intensity maps of diffraction spots. (**g**) TEM image of single-layer graphene edge. (**h**) TEM image of multilayer graphene edge.

**Figure 4 nanomaterials-13-02634-f004:**
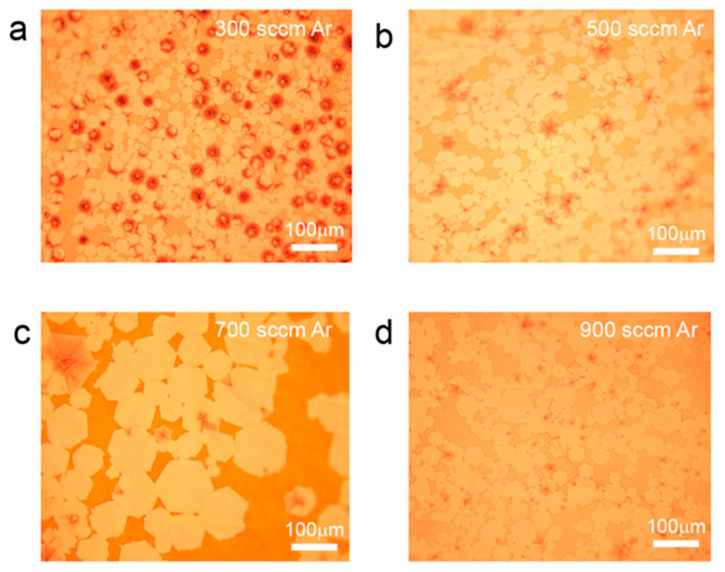
Optical photographs of interlayer-coupled multilayer graphene grown on liquid Cu under different Ar gas flow rates. The growth conditions were set at 100 H_2_ sccm, 0.8 sccm CH_4_ and 300 sccm Ar in (**a**), 500 sccm Ar in (**b**), 700 sccm Ar in (**c**), and 900 sccm Ar in (**d**), respectively.

**Figure 5 nanomaterials-13-02634-f005:**
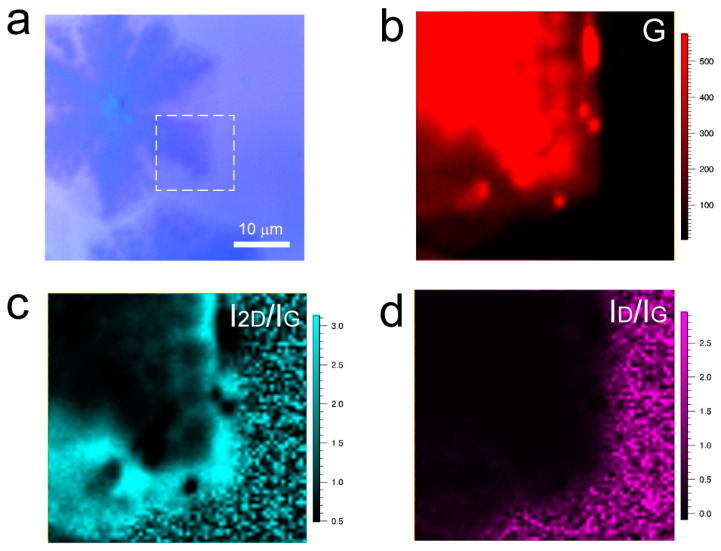
(**a**) Optical photograph of the sample with carbon species continuing to accumulate at the edge of graphene domains, (**b**–**d**) Raman mapping spectra of the G peak and *I*_2D_/*I*_G_ and *I*_D_/*I*_G_ ratios at the position indicated by the white dashed box in Figure (**a**).

**Figure 6 nanomaterials-13-02634-f006:**
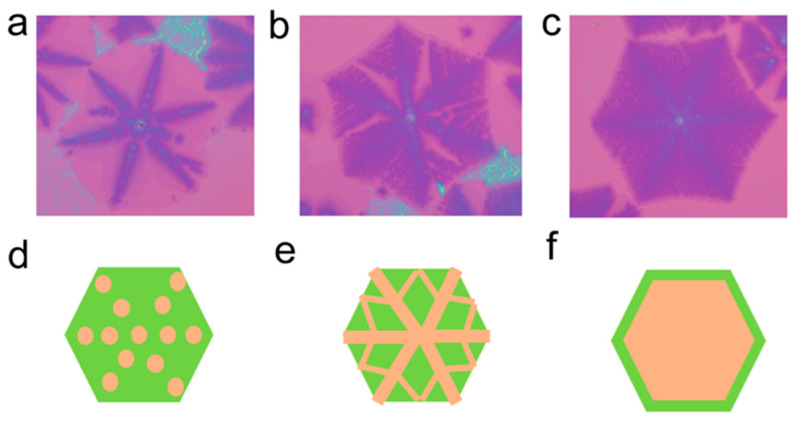
(**a**–**c**) Optical images of umbrella-shaped, interlayer-coupled multilayer graphene at different growth stages and (**d**–**f**) corresponding schematic illustrations of the structure in sequence.

## Data Availability

Data can be available upon request from the authors.

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
