# Peer review of "Controllable Synthesis and Growth Mechanism of Interlayer-Coupled Multilayer Graphene"

_nanomaterials, 2023, doi:10.3390/nano13192634_

Round 1
Reviewer 1 Report
The manuscript reports on the growth mechanism of multilayer graphene which has potential applications such as in electrical conduction and heat conduction, which continue to emerge. However, several challenges need to be overcome. In this work, a simple control strategy for the preparation of multilayer graphene with interlayer coupling on a liquid Cu substrate was developed. By adjusting the flow rate of carrier gas such as methane or acetylene in the thermal CVD system, the effect of fine-controlling carbon source supply was achieved. The authors have proposed carbon diffusion from the edge of the single-layer graphene underneath the layer of graphene and multilayer graphene with interlayer coupling of different shapes. Through a variety of characterization methods, the authors determined the stacked mode of interlayer coupled multilayer graphene conformed to AB stacked. The small multilayer graphene domains stacked under single-layer graphene were first found, and the growth process and growth mechanism of interlayer coupling multilayer graphene with tail fin and umbrella were proposed, respectively. Finally, this study revealed the growth mechanism of multilayer graphene through edge diffusion, paving the way for the controllable preparation of multilayer graphene on a liquid Cu surface. Despite some studies on the growth of multilayer graphene, it requires major revision at both the scientific and editorial levels.
It lacks technical details and a description of the growth mechanism and characterization. Once these deficiencies are taken care of, the paper can be reconsidered.
The English language must be improved.
Author Response
Many thanks for the reviewer’s positive comments on our work. We have polished the manuscript carefully and modified some simple errors. Moreover, we have revised some sentence structures for easy understanding. The corresponding modified parts have been marked in the marked copy of the revised manuscript in red font.
Reviewer 2 Report
Xue et al. present in this manuscript the development of a novel method for synthesizing interlayer-coupled multilayer graphene on a liquid Cu substrate under atmospheric pressure using chemical vapor deposition. The morphology of graphene was characterized using SEM, AFM, and optical microscopy. The layer number and crystallinity were analysed using Raman spectroscopy and TEM. In general, the manuscript is well-written, and the results deserve publication. The content of the manuscript fits the scope of the journal.
Minor comments:
--- Introduction. Please cite the following review at the end of the first sentence (page 1, line 28): Adv. Funct. Mater. 2017, 27, 1702891. DOI: 10.1002/adfm.201702891
--- There are several typing mistakes in the manuscript, for example dot before the literature, missing space between words or brackets. Please correct typos throughout the manuscript.
--- page 10, line 416 (conclusion): “you” must be replaced with “we” or use the following sentence structure: “the "self-limiting" growth mode of single-layer graphene on the Cu surface was disrupted”.
--- References. Authors must follow the reference style of the journal. Please correct.
Only minor editing of English language is required.
Author Response
We appreciate the reviewer’s constructive comments on our work. As the reviewer kindly pointed, we have cited the review (Adv. Funct. Mater. 2017, 27, 1702891. DOI: 10.1002/adfm.201702891) at the end of the first sentence (page 1, line 28). We have corrected the typing mistakes in the manuscript and rechecked the manuscript. Meanwhile, we have corrected the reference style to meet the requirements of the journal. The corresponding changes have been made in the marked copy of the revised manuscript in red font.
Reviewer 3 Report
This submission presents a well-done description and analysis of the growth mode of multilayered graphene and I therefore did not detect any serious scientific issue. This submission can be accepted after some very minor issues listed below.
Main comments:
-References are missing to support the conclusions made line 166
- Legend of figure 3 + all along the text: be careful with the acronym for selected area electron diffraction: SAED and not SEAD
Minor comments
- Abstract, line 11: "such as conduction and heat conduction" ??
- Suppress lines 112 to 114
- Lines 215-216: verb missing
- Line 227: Figure 3c
- Lines 416-417: "self-limiting" growth mode of ....has been disrupted resulting ..."
Author Response
Thanks for the reviewer’s invaluable suggestion. As reviewer advised, we have cited the paper (Proc. Natl. Acad. Sci. 2012, 109, 7992-7996) to support the conclusions made line 166.
We have corrected the acronym for selected area electron diffraction to SAED.
Several typing mistakes on line 11, 215-216, 227 and 416 have been corrected. The content from lines 112 to 114 has been deleted. The corresponding changes have been made in the marked copy of the revised manuscript in red font.
Reviewer 4 Report
I have found this paper interesting. I have few comments for the authors as below:
1-English language needs to get checked.
2-One comparison table comparing this work with previously published ones in the same area is needed before the conclusion section.
3-Detailed applications need to be explained in the discussion section for the future readers of the work.
I have found this paper interesting. I have few comments for the authors as below:
1-English language needs to get checked.
2-One comparison table comparing this work with previously published ones in the same area is needed before the conclusion section.
3-Detailed applications need to be explained in the discussion section for the future readers of the work.
Author Response
Thanks for the reviewer’s constructive advice. We have polished the manuscript carefully and modified some simple errors.
We have added a comparison table comparing this work with previously published ones on supporting information (Table S1 in Supporting Information).
We have added explains for detailed application in discussion section on line 252-255.
The corresponding changes have been made in the marked copy of the revised manuscript in red font.